# Monitoring of Pollutants Content in Bottled and Tap Drinking Water in Italy

**DOI:** 10.3390/molecules27133990

**Published:** 2022-06-21

**Authors:** Giacomo Russo, Sonia Laneri, Ritamaria Di Lorenzo, Ilaria Neri, Irene Dini, Roberto Ciampaglia, Lucia Grumetto

**Affiliations:** 1School of Applied Sciences, Sighthill Campus, Edinburgh Napier University, 9 Sighthill Ct, Edinburgh EH11 4BN, UK; g.russo@napier.ac.uk; 2Department of Pharmacy, School of Medicine and Surgery, University of Naples Federico II, Via D. Montesano, 49, I-80131 Naples, Italy; slaneri@unina.it (S.L.); ritamaria.dilorenzo@unina.it (R.D.L.); ilaria.neri@unina.it (I.N.); irdini@unina.it (I.D.); roberto.ciampaglia@unina.it (R.C.); 3Consorzio Interuniversitario IIstituto Nazionale di Biostrutture e Biosistemi, Viale Medaglie d’Oro, 305, I-00136 Rome, Italy

**Keywords:** drinking water, endocrine disruptors, bisphenols, DEHP, water monitoring, heavy metals

## Abstract

The concentration levels of thirteen organic pollutants and selected heavy metals were investigated in 40 plastics bottled and tap water samples. Some of the selected contaminants have an ascertained or suspected endocrine disrupting activity, such as Bisphenol A (BPA) and its analogs, and Bis 2-ethylhexyl phthalate (DEHP), which are used by industries as plasticizers. The most frequently detected pollutants were Bisphenol AF (BPAF) (detection frequency (DF) = 67.5%, mean 387.21 ng L^−1^), DEHP (DF = 62.5%, mean 46.19 µg L^−1^) and BPA (DF = 60.0%, mean 458.57 ng L^−1^), with higher concentration levels found in tap waters. Furthermore, a possible level of exposure to thirteen pollutants via drinking water intake was calculated. Our findings show that, even though the occurrence of contaminants and heavy metals in drinking waters does not pose an immediate, acute health risk for the population, their levels should be constantly monitored and “hard-wired” into everyday practice. Indeed, the health impact to the continuous and simultaneous intake of a huge variety of xenobiotics from various sources by humans is complex and still not fully understood.

## 1. Introduction

Human water demands depend upon various factors such as climate, physical activity, and diet. However, the average daily water requirement for a sedentary adult’s daily intake, ranges from 1.5 to 2.0 L, besides the water already contained in foodstuff [1]. Therefore, due to the very high quantities needed by the human body, water is considered a macronutrient. Unfortunately, one of the significant sources of human exposure to pollutants is paradoxically drinking water, because of its universal solvent properties of many environmental contaminants. European legislation [2] refers to the natural mineral waters as waters “originated from an aquifer or underground reservoir, spring from one or more natural or bore sources and having specific hygienic features and, eventually, healthy properties” [3]. Natural mineral waters differ from drinking water for their spring purity, constant mineral composition, and the specific effects that these can determine. Unfortunately, chemicals released from industrial discharges, runoff, landfill leachate, and wastewater effluents can spread into the groundwater due to their incomplete removal after wastewater treatment and, therefore, pollute water already from water springs. Among the many anthropogenic pollutants that humans discharge in the environment, endocrine-disrupting chemicals (EDCs) have gained the world’s scientific attention because of their being hormonally active agents, potentially interfering with endocrine systems [4]. EDCs can be both natural and synthetic, and can mimic hormones, causing developmental, reproductive, brain, and immune diseases, as well as other disorders, including cancer [5,6,7]. People can be simultaneously exposed to multiple EDCs, with a synergistic action affecting metabolic processes from early development and organ functions [8,9,10]. People are mostly exposed to EDCs via the intake of foodstuffs and beverages. Indeed, plastic bottles and metal food cans lined with epoxy resins can release free monomers, which have EDC properties, migrating into the food and getting therefore absorbed by the human body [11,12]. This phenomenon may be magnified for foodstuff consumed in high quantities, such as drinking water. Among the huge number of EDCs, Bisphenol A (BPA) and Bis 2-ethylhexyl phthalate (DEHP) are among the most well-characterized EDCs utilized by industries managing plastics material to store beverages, and components of water supply lines. BPA is commonly used in the production of polycarbonates and epoxy resins intended as food contact materials, including tanks for drinking water [13,14,15,16], and DEHP is a plasticizer modulating the flexibility and hardness of plastic materials [17]. DEHP, BPA and/or other monomers can be added to the polymer during its synthesis to tune its mechanical properties, such as flame resistance, color, plasticity, viscosity, and lubricity. Free monomers can migrate into the food, especially if the packaging is stored under conditions that promote this, such as long times, high temperature, or sunlight and UV rays exposition [18]. In addition, every pollutant may already be present in the water at its source due to groundwater contamination. Even if the EDCs presence in water and other beverages is assessed as the lowest among other food categories, the tap or bottled water occurrence can be justified by the possible migration from the coating materials of cans, bottle tops, and water supply pipes [19]. European Plastics Regulation No. 10/2011 [20] authorizes BPA as a monomer in the production of plastic with a specific migration limit of 0.05 mg Kg^−1^ of food and sets DEHP migration limit to a maximum of 1.5 mg Kg^−1^. Despite the sufficiently high limits that make acute intoxication unlikely, the US Environmental Protection Agency (EPA) and US National Toxicology Program (NTP) committees assessed “low dose” effects for several EDCs, defining them “as any biological change occurring in the range of typical human exposures or at doses lower than those tested in traditional toxicology assessments” [21]. Furthermore, due to restrictions set by these laws, industries replaced BPA with compounds having the same structural scaffold, such as bisphenol F (BPF), bisphenol S (BPS), bisphenol A diglycidyl ether (BADGE), bisphenol E (BPE), bisphenol B (BPB), and Bisphenol AF (BPAF), which are currently covered by less stringent or no restrictions at all. These are erroneously considered less toxic than BPA, but they are conversely suspected to be associated with cell toxicity and adverse health effect equal to or even higher than that of the parental compound [22,23,24,25]. The present study aims at determining thirteen organic and seven inorganic contaminants in plastic bottled natural mineral water and in tap water, marketed or collected from running taps in Italy, respectively. The organic chemicals investigated included BPA and several of its analogues mentioned above, but also organic pollutants from other classes such as:2-chlorophenol (2-CP), as s representative of chlorophenols (CPs), banned or controlled by many countries due to its high toxicity and slow biodegradation, but still used as a wood preservative and frequently detected in aquatic and human samples [26,27];4-nonylphenol (4-NP) is a more toxic [28] biodegradation product than its parental compound, nonylphenol ethoxylate (NPEO), a surfactant used in various consumer products as detergents and wetting products to lower interfacial surface tension [29];1,4-dichlorobenzene (DCB), and1,2,4,5-tetrachlorobenzene (TCB) volatile organic compounds mainly used as solvents [30];Triclosan (TCS), an antibacterial agent in consumer products [31,32,33] and well known environmental pollutants, having suspected or assessed estrogenic activity [34].

Furthermore, the monitoring was extended to several inorganic pollutants to include an array of selected heavy metals, i.e., As, Hg, Pb, Cr, Co, Ni, and Cd. Indeed, heavy metal exposure to living organisms may increase the oxidation stress of cells, possibly inducing DNA damage, protein modification, lipid peroxidation, and, for some of them as Cd, there is rising concern about their action on the endocrine system [35]. Due to their high degree of toxicity, Cd, Cr, Pb, and Hg rank among the priority metals that are of public health significance. These metallic contaminants persist in the environment and can accumulate in the body, leading to multiple organ damage and development of cancer [36]. The metals evaluated in this study shall be those operated by the DM 10-02-2015 (Available online: https://www.gazzettaufficiale.it/eli/id/2015/03/02/15A01419/sg (accessed on 20 May 2022)).

## 2. Results

### 2.1. Method Validation

Calibration curves were performed in the range 15.63–250.00 ng mL^−1^ for bisphenol analogs and 4 NP, and in the range 1.25–100 µg mL^−1^ for the different UV detected analytes and a linear R-squared values (r^2^) of all the calibration curves ranged from 0.986 to 1. The sensitivity of the method was supported by LOD parameters, with values ranging from 0.68 to 5.65 ng mL^−1^ for FLD detected analytes and from 0.16 to 7.24 µg mL^−1^ for UV detected analytes. LOQ parameters, ranged from 2.26 to 18.83 ng mL^−1^ for analytes FLD detected and from 0.53 to 24.15 µg mL^−1^ for analytes UV detected. Selectivity, i.e., the ability to discriminate the analyte under study from the occurrence of other chemicals possible interfering with its signal, was assessed by utilizing blank processed samples using Milli-Q^®^ water and any interfering signal was observed. Matrix effect values were found ranging from the lowest value of 1.025 to 1.140. The accuracy of the method using the water matrix was assessed via recovery at low and high spiking concentrations (UV detected chemicals 20–60 µg mL^−1^, and for FLD detected chemicals 1.0–2.0 µg mL^−1^) of all the investigated chemicals resulting in keeping to what previous reported ranging from 75.2 to 112.5%. Method validation was performed according to 2002/657/EC guidelines [37]. These results indicate that this analytical method provides a reliable response regardless of the concentrations utilized. This is straightforward given that the biological matrices studied in [38] are much more complex than the waters assayed in this work. After running of each sample in triplicate, methanol injections did not show carryover.

### 2.2. Real Samples

The water samples, both tap and bottled waters, were checked for chemical and physical parameters and found to be compliant to the current Italian legislation [3,39]. The selected EDCs were detected in 37 out of 40 investigated water samples, while only three samples (i.e., water samples, 9, 10 and 13) did not contain any of the thirteen screened compounds to any detectable extent (Table 1 and Table 2). Among the FLD molecules, BPF was found in 14 water samples (seven tap water and seven bottled waters), BPE in 11 waters (four tap water and seven bottled waters), BPA in 23 waters (10 tap water and 13 bottled waters), BPB in only seven waters (three tap water and four bottled waters), BADGE was detected in 20 waters (11 tap water and nine bottled waters). BPAF was detected in 26 waters (11 tap waters and 15 bottled waters) (Table 1). The mean concentrations were BPF 55.26 ng L^−1^, BPE 130.01 ng L^−1^, BPA 458.57 ng L^−1^, BPB 43.54 ng L^−1^, BPAF 387.21 ng L^−1^, BADGE 353.71 ng L^−1^, and 4-NP 89.91 ng L^−1^, respectively. Regarding UV detected investigated chemicals: BPS was found in six water samples (four tap water and two bottled waters), 2-CP in only two water samples and both at <LOQ level (one tap water and one bottled waters), DCB in 18 waters (six tap waters and 12 bottled waters), TCB in two water samples (one tap water and one bottled water), TCS was detected in five waters (one tap water and four bottled waters), and DEHP in 25 waters (eight tap waters and 17 bottled waters) (Table 2). The mean concentrations were BPS 30.74 µg L^−1^, DCB 61.84 µg L^−1^, TCB 64.07 µg L^−1^, TCS 151.08 µg L^−1^, DEHP 46.19 µg L^−1^. The most commonly chemical detected by FLD was BPAF with a DF of 67.50% along with BPA (DF = 60%), while DEHP was the most frequently screened contaminant among those UV detected with a DF of 62.50%. Mean and median concentrations, detection frequency (%), and concentration range of all the EDCs under our survey are listed in Table 3 (UV and FLD).

### 2.3. Heavy Metals

All heavy metal concentrations in the samples studied were found to be below the national standard limits (DM 10-02-2015 Available online https://www.gazzettaufficiale.it/eli/id/2015/03/02/15A01419/sg (accessed on 20 May 2022) (Table 4 and Table 5). However, the control and monitoring of heavy metals are important, as their presence could cause serious harm to the population, considering the presence of other substances creating a mixture harmful to human health.

### 2.4. Estimated Daily Intake

Regarding human safety, the estimated daily intakes of individual EDCs considered in our study for male and female adults and young people were plotted and shown in Figure 1. The results indicate that the highest values of EDI concern BPA, BPAF, BADGE and TCS for all the gender and age groups, even if adult female group EDI values are slightly lower than the others, due to, probably, their IR lower if compared to IR of male adult, and due to the higher body weight if compared to the youngest categories. Instead, EDI values were found not considerably affected by the age. Furthermore, we also calculated EDI/TDI ratio (Table 6), of BPA, BPAF, BADGE and TCS, which are the chemicals found at the highest concentration values, based on TDI EFSA values (BPA 4 µg/kg BW/day), (BADGE 0.15 mg/Kg BW/day) and (BPAF and TCS 5 mg/Kg BW/day).

## 3. Discussion

BPA, BPAF, and DEHP (Figure 2a,b), were observed in waters samples with a detection frequency of more than 60%, confirming that the observed contamination by these chemicals is due to the anthropogenic activities impacting on the environment. Indeed, packaging industries widely exploit monomers as plasticizers to manufacture food or beverage contact materials [40]. However, these, and other monomers, often leach from packaging. The highest DF and average concentration values of BPAF and its parent compound BPA found in the water samples (Figure 2a), indicate that industries still use the parent chemical, but at the same time increased the exploitation of structurally related bisphenols, probably due to legal restriction set for BPA.

Literature reports aimed at studying the occurrence of EDCs in drinking waters in various countries around the world were used to compare the data achieved in this study [41,42,43,44,45,46,47,48,49,50,51,52]. For most pollutants, we realized that no straightforward comparison on pollutant contamination can be drawn as current data on the drinking waters retailed in most countries is either missing or outdated. Nevertheless, we still believe it is worth comparing our beverage surveillance data with the most representative papers discussing the contamination levels of the main chemicals covered in the present article, in Europe and overseas. BPA mean concentration (458.57 ng L^−1^) is almost ten times higher than those reported by Valcarcel et al. [42] on 30 chemicals in drinking water of Central Spain (51.23 ng L^−1^), and five times higher than that determined by Maggioni et al. [50] performed only on drinking water from public fountains in 35 Italian cities (up to 102 ng L^−1^). However, the concentration values we retrieved are rather similar to those observed in China by Li et al. [43], who detected BPA concentrations up to 317 ng L^−1^. Furthermore, bisphenol analogs in drinking waters were investigated by Zhang and coworkers in source and drinking waters in China [44], with lower concentration values than those reported in our current study.

The 4-NP concentration values we retrieved (89.91 ng L^−1^) were close to those by Maggioni et al. (84 ng L^−1^) [50], but higher than those by Li et al. [43] in China (1.987 ng L^−1^). However, 4-NP concentration in drinking water retailed in Spain by Esteban et al. in Spain (15.0 ng L^−1^) [41], and by Kuch and Ballschmiter [45] in Germany (16.0 ng L^−1^) were about one five times lower than those recorded in the present study.

Triclosan is an antimicrobial agent added to many household and personal care products such as bar soaps, detergents, body, and hair wash. Several studies have reported its occurrence in water samples in concentrations up to 9.74 ng L^−1^, which are much lower than those found in our study [51].

Our results concerning DEHP concentrations showed values in the range 2.10–383.48 µg L^−1^, which is higher than those reported by Jeddi et al. [48] in Iran (0.217 μg L^−1^) and in a literature review by Luo et al. [49], which reporting average levels of DEHP in bottled water retailed in Thailand, Croatia, and the Czech Republic, as high as 61.1, 8.8, and 6.3 μg L^−1^, respectively.

We point out that the daily dietary intakes estimated in our study may underestimate the actual exposures because we considered only water consumption, while many foodstuffs other than drinking water can be contaminated with EDCs and or other chemicals [15,52]. With regard to the comparison between tap and bottled waters, our data support higher concentration levels for each compound (Figure 3) in tap water than bottled water, except for TCS, probably due to the higher purity of source water and the absence of any water treatments.

## 4. Materials and Methods

### 4.1. Analysis of Organic Pollutants

#### 4.1.1. Reagents and Chemicals

For the analysis of organic pollutants, analytical standards, BPF (CAS No.620-92-8), BPS (CAS No.80-09-1), BPA (CAS No.80-05-7), BADGE (CAS No.1675-54-3),2-CP (CAS No.95-57-8), 4-NP (CAS No.104-40-5), DCB (CAS No.106-46-7), TCB (CAS No.95-94-3), DEHP (CAS No.117-81-7), and TCS (CAS No.3380-34-5) were purchased from Sigma-Aldrich (Dorset, United Kingdom). BPE (CAS No.2081-08-5), BPB (CAS No.: 77-40-7), and BPAF (CAS No.1478-61-1) were purchased from TCI Europe (Zwijndrecht, Belgium). Methanol (HPLC analytical grade) and formic acid (minimum purity ≥ 95%) were both purchased from Sigma-Aldrich (Milan, Italy). Milli Q water (from here onward named “distilled water”) was produced in-house, and its conductivity was 0.055 μS cm^−1^ at 25 °C (resistivity equals 18.2 MΩ cm).

#### 4.1.2. Samples and Standard Solutions

All the standard stock solutions (2.0 mg mL^−1^) of each compound were prepared in methanol, as well as those of the mixed standard solution of all the investigated chemicals; the solutions were stored at −24 °C in the dark until the use. The calibration solutions (15.63–250.00 ng mL^−1^ for bisphenols and 4-NP, which were detected by fluorescence detection and 1.25–100.00 μg mL^−1^, for all the other chemicals, which were detected by UV) were freshly prepared for instrumental analysis. Plasticware was treated with a solution of 50:50 n-hexane: tetrahydrofuran [53], water and solvents used for sample preparation were previously analytically [38] verified as chemicals free to avoid any possible background contamination.

From June until December 2019, we retailed 40 water samples, of which 27 drinking water from various Italian wellsprings (North, Central and South of Italy, and one marketed in Italy but from French spring), packaged in bottles of polyethylene terephthalate (PET), one of the most used types of plastic for beverages for its convenience and low cost, and 13 tap water samples from the water distribution Italian system. Figure 4 shows the sampling sites of tap waters and the spring sites of marketed bottled water. For each brand, we took two samples from two different bottles with the same batch number. Tap waters were directly collected from running taps after 2 min of flushing. Each sampling was taken into two pre-rinsed glass bottles, stored at 4 °C, and analyzed within one week.

#### 4.1.3. Sample Preparation

Figure 5 synthetizes samples preparation and analyses steps applied in this study. In brief, the excessive residual chlorine in tap water was removed using 0.2 g L^−1^ of ascorbic acid to fix the concentration of chlorination by products in time [54]. After that, a 200.0 mL volume of each water sample was ultrasonicated (Elmasonic S 30/H, frequency 40 kHz) for 15 min to remove CO_2_ and then filtered through a 0.45 μm nylon membrane. SPE (200-mg Strata X Polymeric Sorbent from Phenomenex, Torrance, CA, USA) was performed on water samples (175 mL) added of 75 mL of methanol to quantitatively solubilize the investigated chemicals. The SPE reversed-phase cartridge were conditioned with 4.0 mL of methanol and activated by 4.0 mL of distilled water according to the SPE manufacturer guidelines. Afterward the sample solutions were loaded onto the cartridge, washed with a total volume of 4.0 mL of a 95/5 (*v*/*v*) water/methanol solution, and subsequently with 12.0 mL of 80/20 (*v*/*v*) water/methanol. Finally, the target contaminants were eluted using 8.0 mL of methanol. All the procedures were vacuum driven. The eluate was evaporated to dryness (Thermo Scientific Savant DNA SpeedVac Concentrator Kits), and the residue was reconstituted with 1.0 mL of acetonitrile for the analysis.

#### 4.1.4. Analysis

A separation chromatographic method, already reported in the literature [38], was fully applied. In brief, each sample was injected three times. Analyses were performed at room temperature (22 ± 2 °C). A high-performance liquid chromatography (LC-20 AD VP; Shimadzu Corp., Kyoto, Japan), equipped with an ultraviolet (UV)–visible detector (Shimadzu Model SPD10 AV) set at λ 220 nm and Fluorescence detector (FLD) set at the excitation wavelength of 263 nm and at the emission wavelength of 305 nm was conducted. The analytical column was a reversed-phase LC column Kinetex phenyl-hexyl (100 Å, 150 × 4.6 mm, 5.0 μm particle size), equipped with a precolumn (4 × 3.0 mm) (Phenomenex, Torrance, CA, USA). All mobile phases were vacuum filtered through 0.45 μm nylon membranes (Millipore, Burlington, MA, USA). BPF, BPE, BPA, BPB, BPAF, BADGE and 4-NP were determined by FLD, while BPS, 2-CP, DCB, TCB, TCS and DEHP were Ultraviolet detected (UV). Data acquisition and integration were accomplished by Cromatoplus 2011 software. Methanol injections were made randomly to assess that no carryover occurred.

### 4.2. Heavy Metals Analysis

#### 4.2.1. Sample Preparation

For the analysis of heavy metals, multi-element standard solution IV 1000 mg L^−1^: Ag, Al, Ba, Bi, Ca, Cd, Co, Cr, Cu, Fe, Ga, In, K, Mg, Mn, Na, Ni, Pb, Sr, Tl, Zn and a Hg and As standard solution of 1000 mg L^−1^,—were purchased from Merck KGaA (Darmstadt, Germany). Water with same specifications as discussed in 4.1.1 was used.

#### 4.2.2. Heavy Metal Analysis

Calibration blank (Cal Blk) and calibration standard (Cal Std) solutions were prepared at concentrations of 0 μg L^−1^ (Cal Blk), 4 μg L^−1^ (Cal Std 1), 12 μg L^−1^ (Cal Std 2), and 20 μg L^−1^ (Cal Std 3), using the selenium standard stock solution, or multi-element standard stock solution, into separate 50 mL DigiTUBE^®^ tubes (SCP SCIENCE, Baie-D’Urfe, QC, Canada) with the addition of 0.5 mL internal standard stock solution, 0.5 mL methanol and 5 mL concentrated nitric acid, and then diluted to the final volume with water. A Shimadzu GFA-EX7i graphite furnace mounted on a Shimadzu model 6300 atomic absorption spectrophotometer was used with a Shimadzu ASC-6100 autosampler. Pyrolytically coated graphite tubes and background correction (D2 lamp) were utilized. Working solutions were prepared by diluting immediately before use standard solutions for absorption a spectroscopy.

### 4.3. Exposure Assessment

The estimated daily intake (EDI ng/kg BW per day) of each chemical under our research, was calculated considering the water consumption, gender, age, and body weight (BW) according to the following Equation reported by Shi et al. [55]:EDI= (C × IR × AP)/BW
where C is the average concentration for each detected chemicals in water sample (ng L^−1^), IR is the ingestion rate of water, AP is the absorption percent of intake, which is assumed to be 100% for drinking water, and BW is body weight (kg). Concentrations below the LOQs were assumed as LODs, to calculate the average values. Age and gender-specific BW along with the intake of drinking water were considered, therefore IR for water and BW were calculated as follows: 2.23 L/day for a 78.4 kg adult male (≥18 years old), 1.65 L/day for a 62.2 kg adult female (≥18 years old), 0.81 L/day for a 31.6 kg boy (0–14 years old), and 0.76 L/day for a 28.73 kg girl (0–14 years old), respectively, and according to literature [55]. For young people (0–14 years old) BW used for EDI calculation was based on the mean value of three different subcategories (0–2.9, 3–9.9 and 10–14 years).

### 4.4. Data Analysis

All statistical analysis was done using a commercially available statistical package for personal computers (Microsoft Excel^®^ 2016, Washington, DC, USA). Data are expressed as the mean of three values.

## 5. Conclusions

The occurrence of organic and inorganic contaminants still raises many questions about risk they may pose to human health. In this pilot monitoring study, several pollutants with suspected or ascertained EDC properties were detected in drinking water at different concentration ranges spanning from ng to µg per L. All forty waters had a total content not exceeding 10 milligrams of total constituents released per dm^2^ of food contact surface (mg/dm^2^) according to the Regulation (EU) No 10/2011 and its amendment Regulation (EU) 2018/213, on plastic materials and articles intended to come into contact with foodstuffs [20].

These findings evidence the need of more frequent surveillance practices aimed at detecting and quality control freshwaters intended for human consumption. However, it is essential to emphasize that the co-occurrence of other contaminants, as well as “natural” elements such as heavy metals, despite their low concentration values, can cause additive/synergistic interactions with possible unexpected effects on human health, especially if the exposure is over a long period of time. Consistently, a proposal to revise the Drinking Water Directive, aimed at updating the water quality standards, including for the first-time EDCs, has been recently approved [56]. This study highlights the importance of assessing their occurrence, regardless of its being low doses, even more in drinking water than other foods. This is because the consumption of water is high and frequent, and assessing the human exposure to contaminants is also highly relevant to inform the legislative work of the main government bodies.

## Figures and Tables

**Figure 1 molecules-27-03990-f001:**
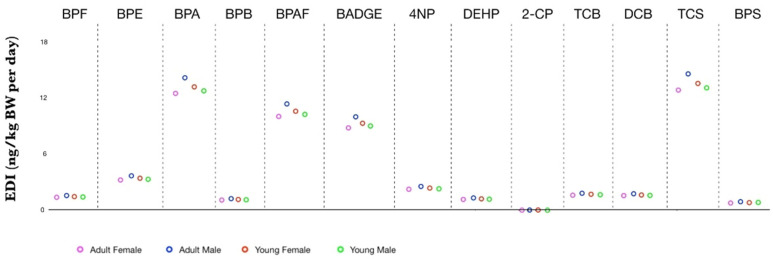
Estimated daily intake (ng/kg BW per day) of detected EDCs via drinking water consumption for age and gender-specific groups.

**Figure 2 molecules-27-03990-f002:**
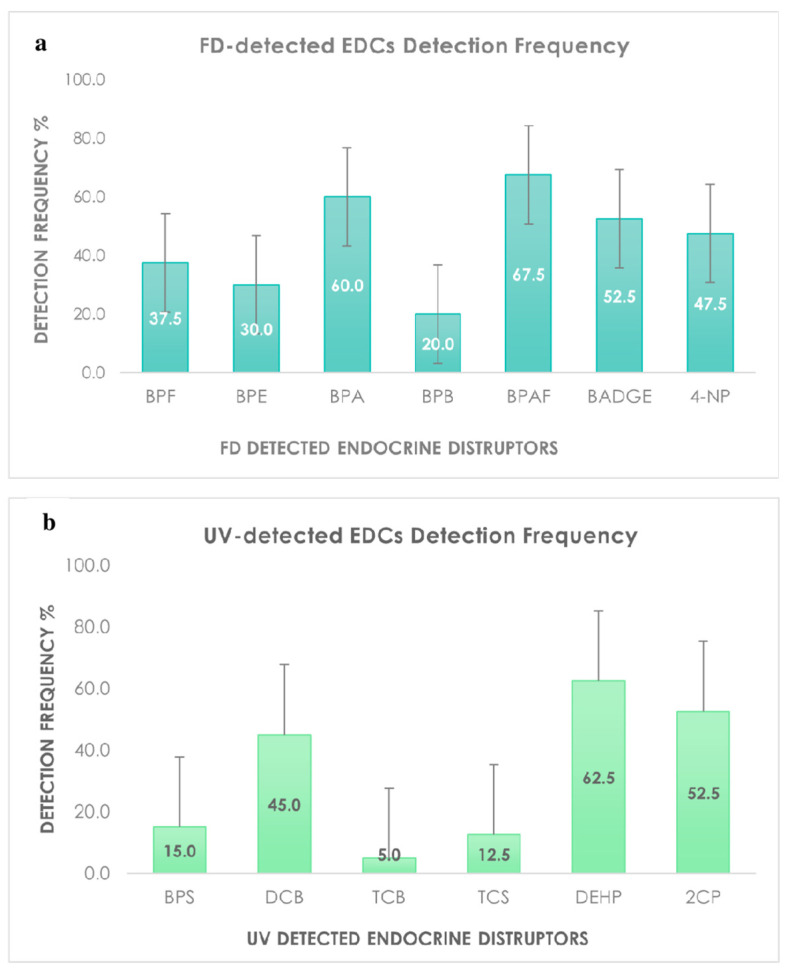
Detection frequency of FD (**a**) and UV (**b**) detected EDCs.

**Figure 3 molecules-27-03990-f003:**
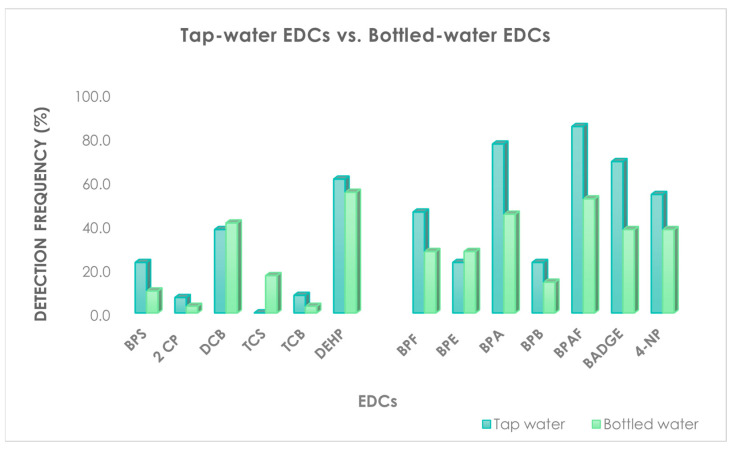
Detection frequency of each investigated chemicals in tap water vs. bottled waters.

**Figure 4 molecules-27-03990-f004:**
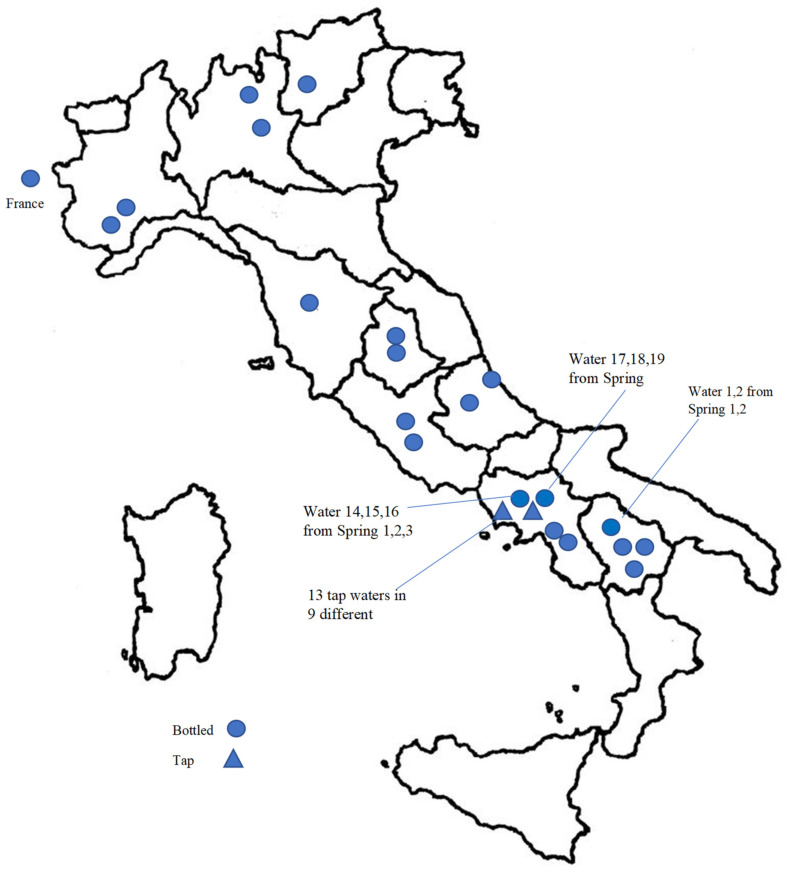
Map of tap water sampling and marketed bottled water springs. Waters 1, 2, 14–16, and 17–19 were of different brand and different spring in the same area; tap waters, were sampled in different cities (Waters 21, 22, 27, 29, 30, 34–36) or in the same city, but served by different aqueducts or pipelines (city 1: Waters 23, 28; city 2 Waters 24–26).

**Figure 5 molecules-27-03990-f005:**
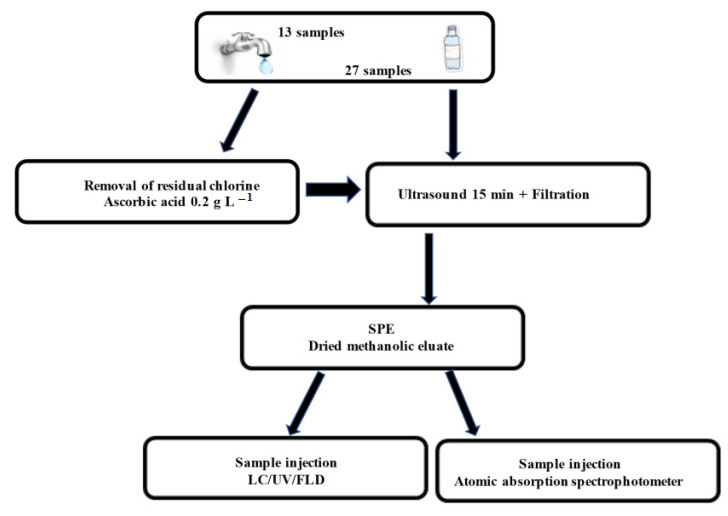
Samples preparation and analysis procedure.

**Table 1 molecules-27-03990-t001:** FLD detected chemicals under investigation.

*Sample*	*Origin*	*BPF*	*BPE*	*BPA*	*BPB*	*BPAF*	*BADGE*	*4-NP*
		ng L^−1^	ng L^−1^	ng L^−1^	ng L^−1^	ng L^−1^	ng L^−1^	ng L^−1^
*Water 1*	South Italy spring 1	*	88.37	*	*	811.61	*	117.95
*Water 2*	South Italy spring 2	*	11.65	*	192.36	512.39	*	81.55
*Water 3*	Central Italy	*	43.11	*	*	1544.68	*	156.71
*Water 4*	South Italy	30.29	*	48.94	*	57.35	23.04	*
*Water 5*	South Italy	9.13	5.63	4.34	*	639.51	29.74	67.81
*Water 6*	North Italy	*	*	*	*	322.37	*	188.09
*Water 7*	North Italy	*	*	*	*	179.95	*	*
*Water 8*	Central Italy	*	*	*	21.78	*	21.15	*
*Water 9*	Central Italy	*	*	*	*	*	*	*
*Water 10*	Central Italy	*	*	*	*	*	*	*
*Water 11*	South Italy	*	*	*	*	110.05	*	*
*Water 12*	North Italy	*	*	50.57	*	*	375.49	*
*Water 13*	Central Italy	*	*	*	*	*	*	*
*Water 14*	South Italy spring 1	*	16.03	*	*	54.57	52.10	41.80
*Water 15*	South Italy spring 2	14.07	9.24	4.34	2.56	62.53	30.96	67.39
*Water 16*	South Italy spring 3	7.96	67.91	28.85	*	863.45	*	97.34
*Water 17*	South Italy spring 1	*	*	*	*	165.17	19.04	33.64
*Water 18*	South Italy spring 2	*	*	*	*	51.58	*	*
*Water 19* *Water 20* *Water 21 #* *Water 22 #* *Water 23 #* *Water 24 #* *Water 25 #* *Water 26 #* *Water 27 #* *Water 28 #* *Water 29 #* *Water 30 #* *Water 31* *Water 32* *Water 33* *Water 34 #* *Water 35 #* *Water 36 #* *Water 37* *Water 38* *Water 39* *Water 40*	South Italy spring 3 Central Italy spring 1 South Italy South Italy South Italy South Italy pickup site1 South Italy pickup site 2 South Italy pickup site 3 South Italy South Italy South Italy South Italy Central Italy South Italy France South Italy South Italy South Italy Central Italy North Italy South Italy Central Italy	42.02 * * 63.05 * * * 327.42 * * * 18.18 * 9.60 6.24 77.29 127.42 11.60 * * * 29.37	* * 23.20 * * * * 164.01 * * * * * * * 779.53 * * * * * 223.3	18.97 * 4.34 27.47 4.34 * * 204.54 26.83 * 46.17 12.82 85.44 30.64 25.44 68.19 329.19 56.69 52.13 716.00 1050.00 900	* * * * 3.90 * 1.16 * * * * * * * * 67.75 * * * 15.24 * *	* 485.90 582.82 866.37 37.71 942.61 1.02 373.44 345.55 35.15 109.59 871.92 19.82 * * 20.30 * * * * * *	* 214.64 49.01 177.72 17.70 101.17 * 75.85 * * 1246.91 * * * * 67.97 2363.46 17.09 * 175.15 173.45 1842.64	136.22 128.09 66.80 34.53 4.39 133.47 * 120.18 * 57.56 * 84.87 * * * * * * * * * *

Values < LOQ = LOD values # Tap waters. * not detected.

**Table 2 molecules-27-03990-t002:** UV detected chemicals under investigation.

*Sample*	*Origin*	*BPS*	*2-CP*	*DCB*	*TCS*	*TCB*	*DEHP*
		µg L^−1^	µg L^−1^	µg L^−1^	µg L^−1^	µg L^−1^	µg L^−1^
*Water 1*	South Italy spring 1	*	*	26.28	*	*	21.51
*Water 2*	South Italy spring 2	6.58	*	*	19.14	*	21.59
*Water 3*	Central Italy	*	*	*	*	*	22.53
*Water 4*	South Italy	0.24	*	1.49	*	*	5.89
*Water 5*	South Italy	*	*	6.76	*	*	5.33
*Water 6*	North Italy	*	*	*	*	*	20.0
*Water 7*	North Italy	*	*	*	*	*	*
*Water 8*	Central Italy	*	*	*	*	*	*
*Water 9*	Central Italy	*	*	*	*	*	*
*Water 10*	Central Italy	*	*	*	*	*	*
*Water 11*	South Italy	*	*	*	*	*	*
*Water 12*	North Italy	*	*	*	*	*	*
*Water 13*	Central Italy	*	*	*	*	*	*
*Water 14*	South Italy spring 1	*	*	15.30	*	*	21.55
*Water 15*	South Italy spring 2	*	*	*	*	*	6.15
*Water 16*	South Italy spring 3	*	0.16	27.22	0.26	*	32.46
*Water 17*	South Italy spring 1	*	*	5.62	*	*	6.20
*Water 18*	South Italy spring 2	*	*	*	*	*	2.10
*Water 19*	South Italy spring 3	*	*	16.29	*	*	*
*Water 20*	Central Italy spring 1	*	*	*	*	*	24.65
*Water 21 #*	South Italy	*	*	20.18	*	*	3.57
*Water 22 #*	South Italy	1.83	*	11.03	*	*	5.06
*Water 23 #*	South Italy	*	*	*	*	*	2.10
*Water 24 #*	South Italy pickup site1	*	*	24.26	*	*	6.95
*Water 25 #*	South Italy pickup site 2	*	*	*	*	*	*
*Water 26 #*	South Italy pickup site3	11.84	0.16	22.18	*	*	35.28
*Water 27 #*	South Italy	*	*	*	*	*	*
*Water 28 #*	South Italy	*	*	*	*	31.03	39.24
*Water 29 #*	South Italy	51.45	*	*	*	*	*
*Water 30 #*	South Italy	*	*	19.05	*	*	5.04
*Water 31*	Central Italy	*	*	*	*	*	*
*Water 32*	South Ital	*	*	207.98	*	*	*
*Water 33*	France	10.04	*	16.51	*	*	34.90
*Water 34 #*	South Italy	*	*	*	162.29	*	383.48
*Water 35 #*	South Italy	*	*	233.40	*	*	*
*Water 36 #*	South Italy	*	*	*	*	*	*
*Water 37*	Central Italy	*	*	112.80	*	*	121.26
*Water 38*	North Italy	*	*	127.47	67.16	97.12	134.52
*Water 39*	South Italy	*	*	*	355.73	*	75.00
*Water 40*	Central Italy	*	*	219.36	*	*	118.46

Values < LOQ = LOD values. # Tap waters. ***** not detected.

**Table 3 molecules-27-03990-t003:** Mean, median, range of concentration, and detection frequency (DF%) of the investigated chemicals. Values are expressed as ng L^−1^ for FLD detected chemicals and as µg L^−1^ for UV detected chemicals. NA = not applicable.

*Compound*	*Mean*	*Median*	*Range*	*DF%*
** *FLD detected* **				
** *BPF* **	55.26	23.78	6.24–327.42	37.5
** *BPE* **	130.01	45.56	2.92–779.53	30.0
** *BPA* **	458.57	46.17	4.34–1050.00	60.0
** *BPB* **	43.54	9.57	1.16–192.36	20.0
** *BPAF* **	387.21	165.17	1.02–1544.68	67.5
** *BADGE* **	353.71	71.91	17.7–2363.46	52.5
** *4-NP* **	89.91	67.81	4.39–188.09	47.5
** *UV detected* **				
** *2-CP* **	NA	NA	NA	5.0
** *BPS* **	30.74	8.31	0.24–51.45	15.0
** *DCB* **	61.84	13.77	1.49–233.4	45.0
** *TCB* **	64.07	NA	31.03–97.12	5.0
** *TCS* **	151.08	19.14	0.26–355.73	12.5
** *DEHP* **	46.19	21.59	2.10–383.48	62.5

**Table 4 molecules-27-03990-t004:** Analytical parameters for the heavy metals screened in the present study.

*Element*	*Wavelength (nm)*	*Slit Width (nm)*	*LOD (mg/L)*	*LMA* *(mg/L) **	*Calibration Curve*	*LR*	*R^2^*
** *Hg* **	253.7	0.2	0.0001	0.0010	y = 0.0005 x + 0.0001	0–20	0.9810
** *Pb* **	283.3	0.2	0.001	0.010	y = 0.0315 x − 0.01454	0–20	0.9990
** *Cr* **	357.9	0.5	0.0001	0.050	y = 0.0008 x + 0.0089	0–10	0.9972
** *Co* **	240.7	0.2	0.0001	NA.	y = 0.0994 x + 0.0279	0–20	0.9609
** *Ni* **	232.0	0.2	0.0001	0.020	y = 0.0754 x + 0.0143	0–20	0.9983
** *Cd* **	228.8	0.5	0.0003	0.003	y = 0.0127 x + 0.00175	0–10	0.9983
** *As* **	193.7	0.2	0.001	0.010	y = 0.0016 x + 0.0006	0–20	0.9797

LMA: Limit maximum admissible. LR: Linear range. NA = not applicable. * DM 10-02-2015.

**Table 5 molecules-27-03990-t005:** Heavy metals analysis in forty investigated water samples.

*Sample*	*Hg*	*Pb*	*Cr*	*Co*	*Ni*	*Cd*	*As*
*Water 1*	<0.0001	0.000223	0.00056	<0.0001	0.0005	0.00056	0.00081
*Water 2*	<0.0001	0.000223	0.00056	<0.0001	0.0005	0.00056	0.00081
*Water 3*	<0.0001	0.000162	0.00015	<0.0001	0.00016	0.00018	0.00018
*Water 4*	<0.0001	0.000555	0.00034	<0.0001	0.00063	0.00084	0.00088
*Water 5*	<0.0001	0.000398	0.00022	<0.0001	0.00068	0.00088	0.00238
*Water 6*	<0.0001	0.00022	0.00088	<0.0001	0.00045	0.00059	0.00281
*Water 7*	<0.0001	0.000162	0.00015	<0.0001	0.00016	0.00018	0.00018
*Water 8*	<0.0001	0.000162	0.00015	<0.0001	0.00016	0.00018	0.00018
*Water 9*	<0.0001	0.000224	0.00056	<0.0001	0.00022	0.00021	0.0033
*Water 10*	<0.0001	0.000433	0.00023	<0.0001	0.00027	0.00034	0.00029
*Water 11*	<0.0001	0.000432	0.00022	<0.0001	0.00033	0.00033	0.00344
*Water 12*	<0.0001	0.000162	0.00015	<0.0001	0.00016	0.00018	0.00018
*Water 13*	<0.0001	0.000311	0.00039	<0.0001	0.00044	0.00022	0.00026
*Water 14*	<0.0001	0.000331	0.00025	<0.0001	0.00361	0.00079	0.00178
*Water 15*	<0.0001	0.000313	0.00061	<0.0001	0.00264	0.00079	0.00105
*Water 16*	<0.0001	0.000422	0.00038	<0.0001	0.00033	0.00085	0.00109
*Water 17*	<0.0001	0.00038	0.00024	<0.0001	0.00048	0.00092	0.00077
*Water 18*	<0.0001	0.000231	0.00034	<0.0001	0.00019	0.00018	0.00055
*Water 19*	<0.0001	0.000231	0.00034	<0.0001	0.00019	0.00018	0.00055
*Water 20*	<0.0001	0.000299	0.00021	<0.0001	0.00056	0.00036	0.00018
*Water 21*	<0.0001	0.000561	0.00027	<0.0001	0.00244	0.00085	0.00126
*Water 22*	<0.0001	0.000259	0.00015	<0.0001	0.00054	0.00081	0.00091
*Water 23*	<0.0001	0.00047	0.00044	<0.0001	0.00066	0.00077	0.00158
*Water 24*	<0.0001	0.000404	0.00017	<0.0001	0.00033	0.00085	0.00084
*Water 25*	<0.0001	0.000404	0.00017	<0.0001	0.00033	0.00085	0.00084
*Water 26*	<0.0001	0.000404	0.00017	<0.0001	0.00033	0.00085	0.00084
*Water 27*	<0.0001	0.000368	0.00023	<0.0001	0.00079	0.00083	0.00109
*Water 28*	<0.0001	0.00047	0.00044	<0.0001	0.00066	0.00077	0.00158
*Water 29*	<0.0001	0.000563	0.00055	<0.0001	0.00019	0.00026	0.00054
*Water 30*	<0.0001	0.000579	0.00046	<0.0001	0.00175	0.00088	0.00103
*Water 31*	<0.0001	0.000299	0.00021	<0.0001	0.00056	0.00036	0.00018
*Water 32*	<0.0001	0.000311	0.00034	<0.0001	0.00041	0.00045	0.0019
*Water 33*	<0.0001	0.000162	0.00015	<0.0001	0.00016	0.00018	0.00018
*Water 34*	<0.0001	0.000579	0.00046	<0.0001	0.00175	0.00088	0.00103
*Water 35*	<0.0001	0.000579	0.00046	<0.0001	0.00175	0.00088	0.00103
*Water 36*	<0.0001	0.000579	0.00046	<0.0001	0.00175	0.00088	0.00103
*Water 37*	<0.0001	0.000341	0.00067	<0.0001	0.00016	0.00032	0.00034
*Water 38*	<0.0001	0.000162	0.00015	<0.0001	0.00016	0.00018	0.00018
*Water 39*	<0.0001	0.000162	0.00025	<0.0001	0.0002	0.00083	0.00077
*Water 40*	<0.0001	0.000456	0.00031	<0.0001	0.00071	0.00067	0.00067

LMA: Limit maximum admissible; LMD: Limit minimum of detection. LMA Hg = 0.001 for all water samples; LMA Pb, As = 0.01 for all water samples; LMA Cr = 0.05 for all water samples; LMA Ni = 0.02 for all water samples; LMA Cd = 0.003 for all water samples; LMA Co = ND for all water samples. LMD Hg, Cr, Co, Ni = 0.0001 for all water samples; LMD Pb, As = 0.001; LMD Cd = 0.0003 for all water samples.

**Table 6 molecules-27-03990-t006:** Calculated EDI/TDI ratio for adult and young, male, and female, based on TDI EFSA values (BPA 4 µg/kg BW/day), (BADGE 0.15 mg/kg BW/day) and BPAF and TCS (5 mg/kg BW/day).

*Analyte*	*Adult Male*	*Adult Female*	*Young Male*	*Young Female*
** *BPA* **	0.003568	0.003147	0.003216	0.003319
** *BPAF* **	2.29 × 10^−6^	2.02 × 10^−6^	2.06 × 10^−6^	2.13 × 10^−6^
** *BADGE* **	6.71 × 10^−5^	5.91 × 10^−5^	6.04 × 10^−5^	6.24 × 10^−5^
** *TCS* **	2.94 × 10^−6^	2.59 × 10^−6^	2.65 × 10^−6^	2.73 × 10^−6^

## Data Availability

Not applicable.

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
