# Peer review of "Monitoring of Pollutants Content in Bottled and Tap Drinking Water in Italy"

_molecules, 2022, doi:10.3390/molecules27133990_

Round 1

Reviewer 1 Report

Review: Monitoring of pollutants content in bottled and tap drinking water in Italy

General comments:

This study analyzed endocrine disrupting micropollutants and some trace metals in bottled water and tap water in Italy and evaluated the exposure levels from the co-occurring contaminants. The occurrence of different emerging contaminants in drinking water is an important theme and of wide interest. The study contributes to the larger body of knowledge on emerging contaminants in drinking water and the exposure assessment for multiple EDCs is also of interest.

 The main concern is with the brevity of the methods and how representative the samples were, especially when comparing tap vs bottled water. Information is missing about the source of the waters. It is understood that the name of the company or location should not be stated, but the number of different bottled water companies (or were all the bottles from the same company?) and the number of different tap locations (these should not all be in the same building?) should be stated. If they are from only one or two companies or buildings (for tap), then that should be stated as a limitation. In that case, the study may only be representative of some locations.

Another concern is whether the chlorine treatment (adding ascorbic acid and ultrasonication) had any impact on the contaminant concentrations. Were there any controls (ultrapure water processed with the same chlorine removal procedure)? This step could have introduced some contaminants, and any efforts to reduce this contamination source should be stated. Also the SPE cartridges could introduce some chemicals. Or were process blanks subtracted to remove any background chemicals? This should be stated in the Methods.

For the comparison of tap water and bottled water, which is a very interesting part of the study, there should be more discussion. The discussion is limited to one sentence (L301-304). There must have been other studies comparing tap water and bottled water. The findings of this study should be placed into the larger context (cite other studies). Also, it seems that even a simple student t-test (unpaired) could provide a measure of the statistical significance. Tap water almost always had a higher concentration of the chemicals of interest. Here it would be important to explain that the tap water was from different taps (if it was) and so for this representative sampling, tap water may have more sources of contamination. The authors allude to the bottled water having higher purity sources and absence of water treatment. Is that known? Perhaps the authors could instead (or in addition) consider the type of water treatment and where the chemicals could originate from (some of this [eg., pipes] is mentioned in the intro, but could be restated in the discussion, where it applies).

For all tables, I suggest moving the special notes below the table, as a footnote (this refers to the notes about LMA, LMD, etc)

Figures with horizontal grid lines are quite busy. In Figs 2a and 2b, these can easily be removed because the value is shown right above the bars. In Figure 3, the lines also may not really be necessary. Boxes around “a” and “b” also add to the difficulty viewing these figures. Overall, the figures could be improved and look more consistent.

There are some inconsistencies in editing. For example superscripts are sometimes used, sometimes missing (eg., ng L-1). These and other editing errors should be corrected.

Detailed comments:

L56: For this passage “such as indeed drinking water”, remove “indeed”.

L216 – 217: It is interesting that some samples did not have any FLD- or UV-detected chemicals. Were more than one bottle from the same company tested and so some had the chemicals and some did not? Or were all the bottles from different companies? And is that why some did not have any chemicals? More information should be given about the samples and discussed here (of course without giving the names of the companies).

L267-269: there should not be only one sentence in a paragraph. Add a sentence of merge this into another paragraph.

L301: this should be a new paragraph comparing tap and bottled water. More information should be added and some additional studies should be cited and discussed.

Reviewer 2 Report

The publication is interesting and probably will be interesting to readers. It deserves to be published in Molecules, but some aspects of the paper need refinement:

1.section 2.1 does not at all present a validation of the method, nor is it in the materials. This needs to be supplemented with concrete data rather than a laconic comment that everything was fine. This is a scientific publication, the results presented therein cannot raise any doubts as to their credibility or quality, therefore, the authors must present step by step the exact and complete results of the validation performed.

2. the methods used by the authors, which are crucial for the results obtained (on which the whole publication is based), should be discussed. Please present a few chromatograms, one for the mixed standards and one for the extracts. Nowhere are any specific conditions mentioned as to the mobile phases used. What were they? What gradient? Please be sure to provide more detailed results for SPE. What were the recoveries? How did the authors investigate the breakthrough volume of the deposit (for only 200ml of adsorbent material they introduced very large volumes of water - 175ml, so how did the authors see that there was no breakthrough)?

without providing detailed results for analytical methods, the entire publication (and indeed its results) can be easily questioned.

3. What were the values of LOD, LOQ, what was the range of concentrations on the standard curve, what was the matrix effect (these data were presented only for heavy metals, not for other compounds)?

4. Did the determined contents of e.g. bisphenols exceed acceptable standards?

Round 2

Reviewer 2 Report

Paper may be accepted.